# Alanine aminotransferase controls seed dormancy in barley

Kazuhiro Sato[1], Miki Yamane[1], Nami Yamaji[1], Hiroyuki Kanamori[2], Akemi Tagiri[2], Julian G. Schwerdt[3], Geoffrey B. Fincher[3], Takashi Matsumoto[2], Kazuyoshi Takeda[1] & Takao Komatsuda[2]

Dormancy allows wild barley grains to survive dry summers in the Near East. After domestication, barley was selected for shorter dormancy periods. Here we isolate the major seed dormancy gene qsd1 from wild barley, which encodes an alanine aminotransferase (AlaAT). The seed dormancy gene is expressed specifically in the embryo. The AlaAT isoenzymes encoded by the long and short dormancy alleles differ in a single amino acid residue. The reduced dormancy allele Qsd1 evolved from barleys that were first domesticated in the southern Levant and had the long dormancy qsd1 allele that can be traced back to wild barleys. The reduced dormancy mutation likely contributed to the enhanced performance of barley in industrial applications such as beer and whisky production, which involve controlled germination. In contrast, the long dormancy allele might be used to control pre-harvest sprouting in higher rainfall areas to enhance global adaptation of barley.

[1] Institute of Plant Science and Resources, Okayama University, 2-20-1, Chuo, Kurashiki, Okayama 710-0046, Japan. [2] National Institute of Agrobiological Sciences, Tsukuba 305-8602, Japan. [3] ARC Centre of Excellence in Plant Cell Walls, School of Agriculture, Food and Wine, University of Adelaide, Waite Campus, Glen Osmond, South Australia 5064, Australia. Correspondence and requests for materials should be addressed to K.S. (email: kazsato@rib.okayama-u.ac.jp).

Barley (*Hordeum vulgare*) was domesticated *ca.* 10,000 years ago from its wild progenitor *H. vulgare* subsp. *spontaneum* in the Fertile Crescent[1,2]. Wild barley is characterized by long grain dormancy, which lasts for several months after grain maturation[3]. Dormancy can be defined as the failure of an intact, viable seed to germinate under conditions that would normally be favourable for germination[4]. The wild barley's long dormancy means that, initially, the grain will not germinate in response to transient moisture availability and will therefore survive hot, dry summers. However, after some months of higher temperatures in the summer, grain dormancy is broken and this allows the grains to germinate after rains in the autumn. The selection pressure for shortened seed dormancy has been driven by the post domestication sowing activities of barley[5]. The process of malting barley kernels for brewing beer or distilling whisky involves controlled germination. Pedigrees of cultivars show that during the 19th century, malting barleys from the UK and Moravia spread to neighbouring countries and also to East Asia and the Americas[6]. Long periods of seed dormancy became undesirable because they required a longer storage period prior to malting. Thus, the control of dormancy is of great practical importance in agricultural production of crop species and in grain utilization by food and beverage industries.

Two major quantitative trait loci (QTLs) on chromosome 5H, one near the centromere (*Qsd1*) and another in the telomeric region of the long arm (*Qsd2*) have been identified[7]. Of the two QTLs, *Qsd2* was isolated and proved to encode a mitogen-activated protein kinase kinase 3 (ref. 8). The *Qsd1* QTL mapped in the cross of wild barley H602 and malting barley cv. Haruna Nijo[9]. Haruna Nijo carries a dominant short dormancy allele whereas the wild barley carries a recessive long dormancy allele. There are some reports of other genes that control seed dormancy in wheat[10] and rice[11,12], but none are orthologous with *Qsd1*. Furthermore, seed dormancy in wild barley is much longer than seed dormancy observed in these other cereal species.

We have identified a gene that defines a major dormancy trait in wild ancestral forms of barley and show that it encodes alanine aminotransferase, which has not previously been linked with dormancy in plants. We show that a single amino acid substitution in this enzyme reduces the dormancy period of mature barley grain. Our data suggest that the *qsd1* gene for long dormancy appears to be a mutable gene that is used for adaptation to different environments and this is consistent with the large variation in *qsd1* sequence that exists in nature.

## Results

**Qsd1 encodes alanine aminotransferase.** For the initial map-based cloning of *Qsd1*, we identified a genetic substitution line carrying the long dormancy *qsd1* allele of H602, together with Haruna Nijo's short dormancy alleles for the other three QTLs[13,14]. A total of 910 $F_2$ plants were produced to genetically map *Qsd1* (ref. 14). The flanking *EST1* and *EST12* markers (7.4 cM interval, Supplementary Table 1) were used to select recombinants. All 13 expressed sequence tag (EST) markers were assayed in the recombinants. *EST4R* co-segregated with *Qsd1* in a total of 4,792 $F_3$ plants (Fig. 1a). The *EST4R* marker was used for clone selection from genomic libraries of Haruna Nijo[15] and H602 (ref. 2). One and three clones were selected for Haruna Nijo and H602 libraries, respectively. Of these, one clone from each library comprised a contig. An alignment of the genomic sequences from the two genotypes showed that these contigs shared 100,231 bp of sequence. On these contigs a full-length complementary DNA clone[16] *LC054176* maps at the *EST4R* marker and another full-length cDNA[17] *AK372829* maps, 345 bp away, at the *EST5* marker next to *EST4R*. Among the 4,792 $F_3$ plants, one plant $F_3$ #2,253, recombined between *Qsd1* and *EST5*

and another plant $F_3$ #2,372 recombined between *Qsd1* and *EST4F* (Fig. 1a), indicating that *Qsd1* is located in a 9,467 bp region between *EST5* and *EST4F* and thereby *Qsd1* is located in either *LC054176*, *AK372829* or the intergenic region. Homozygous $F_4$ plants from $F_3$ #2,372 discretely segregated *qsd1/qsd1* (long dormancy) and *Qsd1/Qsd1* (short dormancy) in a Mendelian fashion (Fig. 1b).

Within the 9,467 bp region, there were only four non-synonymous single nucleotide polymorphisms (SNPs) and all of them were located in the coding region of *AK372829* (Fig. 1c, Supplementary Table 2), indicating that one or more of these were likely to be responsible for the *Qsd1* phenotype. The *AK372829* gene encodes an alanine aminotransferase (AlaAT), a protein of 495 amino acid residues (Supplementary Fig. 1). For proof-of-function experiments, the AlaAT gene was used to transform barley. However, the Golden Promise variety used in our standard *Agrobacterium*-mediated transformation experiments carries the dominant short dormancy *Qsd1* allele, so we developed knockdown plants by RNA interference (RNAi) with the pANDA vector[18] (Supplementary Fig. 2) with the objective of developing long dormancy lines. Figure 2a shows the effects of *Qsd1* RNAi knockdown on seed germination. Of the two events, #7 $T_2$ transgenic homozygous positive plants showed < 5% germination after 21 days at 21 °C (Fig. 2a,b), while heterozygous plants showed intermediate levels of about 40% germination under the same conditions. These values may be the compared with values of 80–90% obtained for negative control lines and the wild type Golden Promise. Both the homozygous and heterozygous transgenic $T_2$ plants showed decreased transcript levels for the *AK372829* AlaAT gene (Fig. 2c).

We also conducted complementation tests using genetic substitution lines that contained the recessive long dormancy *qsd1* allele of wild barley H602 (Supplementary Fig. 3) in a background of Golden Promise (id. B1H602GP20-7) by transforming with a construct carrying the cloned Haruna Nijo short dormancy *Qsd1* genomic DNA (Supplementary Fig. 4). Figure 2d shows the effects of the transgene *Qsd1* on seed germination. The $T_1$ plants carrying the *Qsd1* transgene showed vigorous seed germination, which was significantly higher than both the transgene negative plants and the parent substitution line (*qsd1/qsd1*) (Fig. 2d,e). Transcript levels in the embryos increased in transgene positive $T_1$ plants (Fig. 2f). These transgenic data clearly demonstrated that the *Qsd1* gene is responsible for shortened dormancy in barley grain. Given the complexity of the mechanisms of seed dormancy and germination and their operation at many levels in the plant[19,20], the large effect of the *Qsd1* gene on grain dormancy in barley indicates that the encoded AlaAT enzyme plays a central role in the release of dormancy in this species.

**Association between SNPs and dormancy.** To identify any association between dormancy and the four SNPs identified in the long and short dormancy alleles of the AlaAT gene, the additive effects of individual *Qsd1* alleles were compared with the parents used for *Qsd1* mapping[9] (Supplementary Tables 2 and 3). Among the four non-synonymous SNPs in the delimited region, SNP E9 matched the *Qsd1/qsd1* alleles. This SNP is associated with a change from a leucine (L) residue at amino acid position 214 in wild barley H602 to a phenylalanine (F) residue in cultivar Haruna Nijo (Fig. 1c). The SNPs E11 and E14 showed a much lower level of correlation with the dormancy phenotype, but it is not possible to exclude the possibility that they also have effects on the dormancy level observed (Supplementary Table 3).

To further confirm that the SNP alleles were correlated with dormancy in barley grain more generally, the germination scores of 353 cultivated and 14 wild barley accessions were analysed.

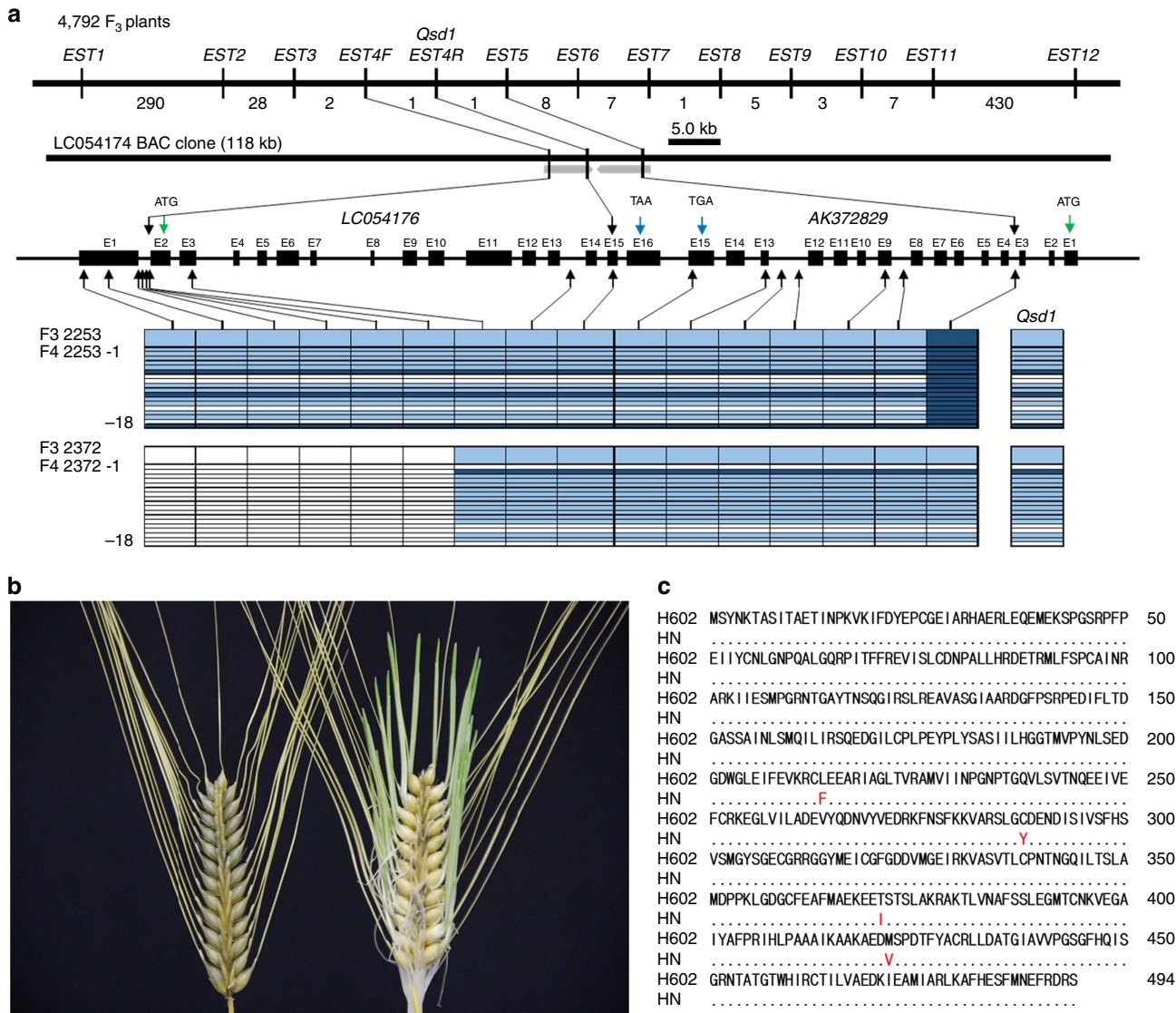

**Figure 1 | The isolation of Qsd1.** (**a**) Mapping of *Qsd1* (QTL for seed dormancy) on the high-density linkage map of Haruna Nijo (less-dormant, *Qsd1*)/H602(dormant, *qsd1*) (ref. 13) and high-resolution mapping of the recombinant chromosome substitution line which includes substituted segment of *Qsd1* and excludes other QTL regions[14]. A Haruna Nijo BAC clone LC054174 (ref. 15) was selected by *EST4R* and read the sequence of gene structures for *EST4R* and *EST5*. Integers are number of recombinants between neighbouring markers. Polymorphisms between Haruna Nijo and H602 (arrows) are used to genotype recombined F3 plants and their offsprings. *Qsd1* was delimited to the heterozygous region in two F3 plants. Alleles are indicated by dark blue (H602), blue (heterozygote), and open (Haruna Nijo). (**b**) Effects of *qsd1* (left) and *Qsd1* (right) alleles on seed germination after five weeks of 25 °C dormancy reduction treatment in F4 near isogenic plants. (**c**) Comparison of amino acid sequences of *Qsd1* between Haruana Nijo (HN) and H602. Non-synonymous amino acids in Haruna Nijo are indicated by red.

The germination (low-dormancy) scores were compared among haplotypes of non-synonymous SNPs in *AK372829* (Supplementary Table 4). Average germination scores of cultivated barleys were lowest in haplotype TACC (non-synonymous SNP haplotype at E14, E13, E11, E9) and highest in CATG (Haruna Nijo haplotype). There was a clear difference in average germination between E9 alleles. Variation at the other three SNP alleles showed relatively smaller effects on germination. The number of haplotypes having allele G in E9 ($n = 46$) was much lower than haplotypes having allele C ($n = 307$). Most of the barleys having the allele G have pedigrees that track malting and brewing applications[6], indicating that the allele G might have resulted from the specific selection of reduced dormancy for the malting process.

To track the origin of the *Qsd1* mutation that resulted in the reduced dormancy time, the genomic regions of *Qsd1* in 27 wild and 210 cultivated barley accessions were sequenced (Supplementary Fig. 5, Supplementary Table 5 and Supplementary Data 1) and aligned (Supplementary Table 6). Eight sub-clusters (clades) of two wild barleys (W), three cultivated types (C) and mixtures of wild and cultivated types (WC) were evident (Fig. 3).

**Structure and diversity of QSD1 protein.** The question now arises as to how the AlaAT enzyme might be a major determinant of the length of barley grain dormancy. There are five AlaAT genes in barley (Supplementary Table 7). The short dormancy gene *Qsd1* is located on chromosome 5HL. *Hv2* is located on chromosome 2HS, while *Hv3* is likely to be on chromosome 1H, *Hv4* is on chromosome 5HS and *Hv5* is on chromosome 2HL (ref. 21). Estimated domains of AlaAT class I and class II

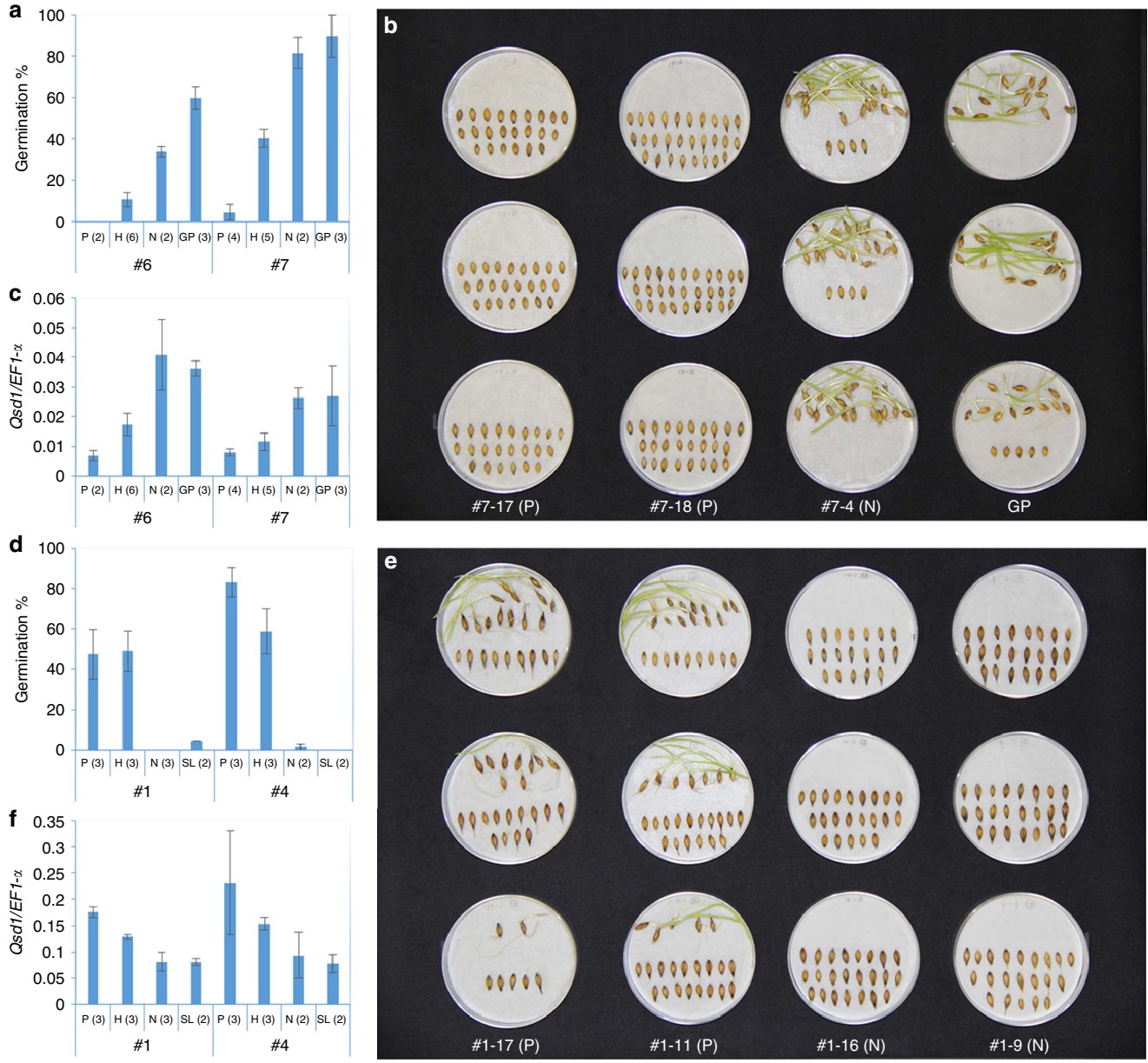

**Figure 2 | Effects of RNAi knockdown and complementation for *Qsd1*.** We have conducted two sets of RNAi events and two sets of complementation events and the effect was consistent in all transgenic lines. The number in parentheses indicates the number of plants. Error bars represent s.e.'s when $n \geq 3$ or data points when $n = 2$. (**a**) Germination after 21 days of 25 °C treatment for dormancy reduction on $T_1$ RNAi heterozygous plants (#6 and #7) derived $T_2$ transgenic homozygous positive (P), heterozygous (H) and negative (N) plants and transformation efficient non-dormant cv. Golden Promise (GP). (**b**) Germination for selected $T_2$ transgenic positive, negative plants and GP. Each Petri dish includes a seed sample from a spike. (**c**) Relative gene expression of *Qsd1* transgene plants to *EF1-α* on the embryo at 28 days after flowering (Supplementary Fig. 10). (**d**) Germination of *Qsd1* transgene plants on the transformation efficient substitution line (SL: id B1H602GP20-7) which has a segment with *qsd1/qsd1* from dormant H602. After 35 days of 25 °C treatment of dormancy reduction, $T_1$ transgene homozygous positive, heterozygous and negative plants derived from heterozygous $T_0$ plants (#1 and #4) show germination or non-germination. (**e**) Germination of *Qsd1* transgene plants on SL. After 35 days of 25 °C dormancy reduction treatment, transgene homozygous positive and negative plants show germination and non-germination, respectively. (**f**) Relative expression levels in *Qsd1* transgene plants of *EF1-α* in the embryo at 28 days after flowering (Supplementary Fig. 10).

subfamilies (IPR004839) are highly conserved. The amino acid sequence of QSD1 = Hv1 are highly similar (*E*-value $< E - 145$) for AlaATs in other plant species and for four barley sequences (Hv2, Hv3, Hv4 and Hv5part) of Haruna Nijo (Fig. 4a,b). None of these sequences have been previously reported to be related to dormancy. Motif analysis of QSD1 and its homologues using the SALAD database[22] showed that Hv3 and Hv4 shared the same motifs with QSD1 (Fig. 4b). One of the non-synonymous amino acid changes from leucine (L) of *qsd1* to phenylalanine (F) of

*Qsd1*, corresponding to E9, was located on motif 11 (Fig. 4c). QSD1 has the only haplotype that shows phenylalanine (F) at E9 among the sequences examined, whereas L is the most common in other species. The result implies that the evolution of *Qsd1* represented the dominant mutation for reduced dormancy in barley.

**Qsd1 expression profiles.** In addressing the question posed above regarding AlaAT function in dormancy, transcriptional profiling

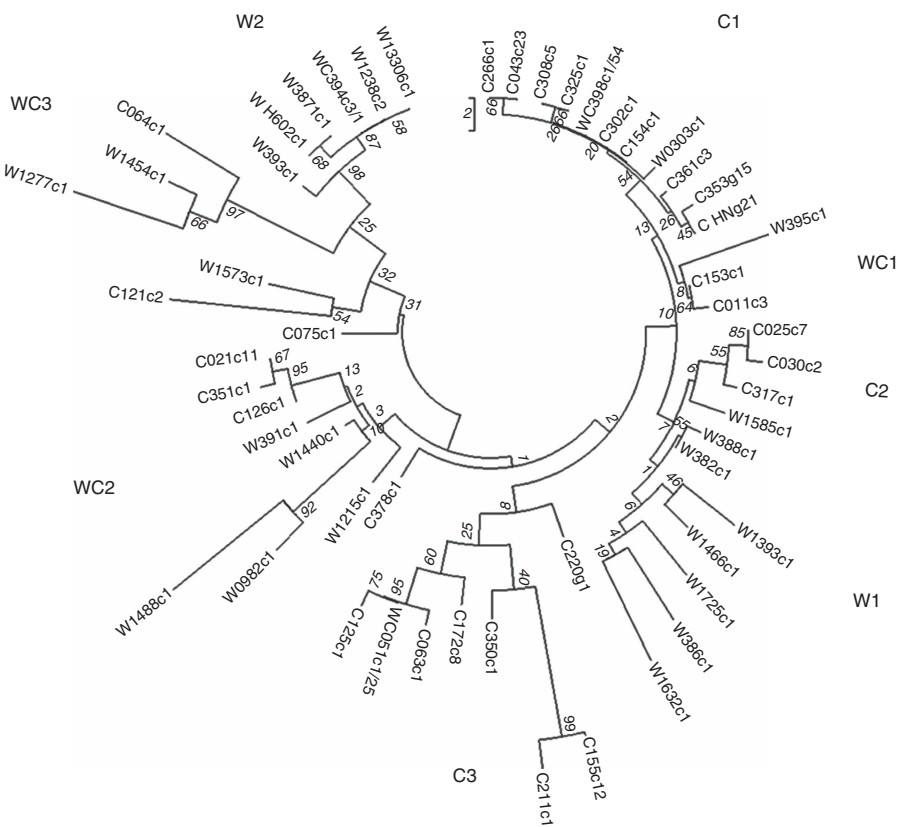

**Figure 3 | Molecular relationships of haplotypes for *Qsd1* in wild and cultivated barleys.** W and C represent wild and cultivated barleys, respectively, and WC means mixtures of wild and cultivated barleys. Haplotype name is composed of type of barley (W, C or WC), representative accession number, non-synonymous SNP genotype at exon 9 (Supplementary Table 6) and number of accessions in the haplotype (order of W/C in case of mixture). Numerals in italic show test values for 1,000 bootstrap replications.

was used to first define the timing of the appearance of enzyme in the grain. The transcription of *Qsd1/qsd1* was detected in developing grains at 21, 28 and 35 days after flowering (Fig. 5a) but transcript levels were very low in grains at 0, 7 and 14 days after flowering. Very low transcript levels were also detected in young or older leaves and roots. The grain-specific expression in the long dormancy haplotype (*qsd1*: H602) was further analysed by RNA *in situ* hybridization 19 days after flowering. Transcripts were detected in the embryo but not in the endosperm (Fig. 5b). The microarray-based analysis of the *Qsd1* homologue in rice also showed that transcription was embro-specific[23] (Supplementary Fig. 6a), which is quite different from the wider expression patterns of paralogous rice genes (Supplementary Fig. 6). The embryo-specific expression in *Qsd1* and *Os1* increases during grain maturation, whereas expression levels of other (*Os2-Os5*, Fig. 4a and Supplementary Fig. 6) rice genes in the embryo decrease during the maturation process. It was possible that the phenotype is caused by temporal differences in expression of the two genes, but no major temporal differences could be detected between AlaAT transcript abundance in Haruna Nijo and H602 (Fig. 5a).

**Structure and diversity of QSD1 protein.** Next, we investigated the possibility that the L214F substitution in short dormancy barleys might affect AlaAT activity. To identify the spatial disposition of L214, we constructed a homology model based on the crystal structure of Hv3 (ref. 24), which is a dimeric, pyridoxal 5′-phosphate-dependent enzyme. Thus, the Hv3 crystal structure has been used here to locate the position of the L214F substitution that defines the difference between the enzymes encoded by the

wild type *qsd1* (L) and mutant *Qsd1* (F) alleles (Supplementary Fig. 7), where L is conserved in most cereal orthologs (Fig. 4c). The location of L214 is shown in Supplementary Fig. 7 but it is not close to the active site, to the cofactor binding site or to the junction zone of the two proteins of the dimer. The L214F substitution has not yet been observed in other plant species (Fig. 4c). So, at this stage one would suspect that the conservative L214F substitution would not have a dramatic effect on enzymatic activity; we believe it is more likely that the region of the substitution is involved in homo-dimer or -trimer formation, or in binding of another, regulatory protein onto the surface of the AlaAT enzyme.

## Discussion

The fact that the qsd1 protein is encoded by the *qsd1* allele in wild barley implies that qsd1 has an essential function in seed dormancy, through a mechanism that is potentially common with other plant species. The qsd1 enzyme has a L residue at amino acid site 214 and this is conserved in orthologous cereal proteins (motif 11, Fig. 4c). The re-sequencing of wild and cultivated barley germplasm representing the global genetic diversity of barley did not show a single accession carrying the lesion allele without qsd1 function. This strongly supports the notion that qsd1, but not QSD1, represents the fundamental function of seed dormancy.

The haplotype analyses (Supplementary Tables 4 and 6) showed that QSD1 protein structures may vary, even among wild barleys. The wild barley H602 (Fig. 3, W_H602: W2 clade) and the cultivar Haruna Nijo (C_HN: C1 clade) were located most distantly among the haplotypes. Of the haplotypes observed

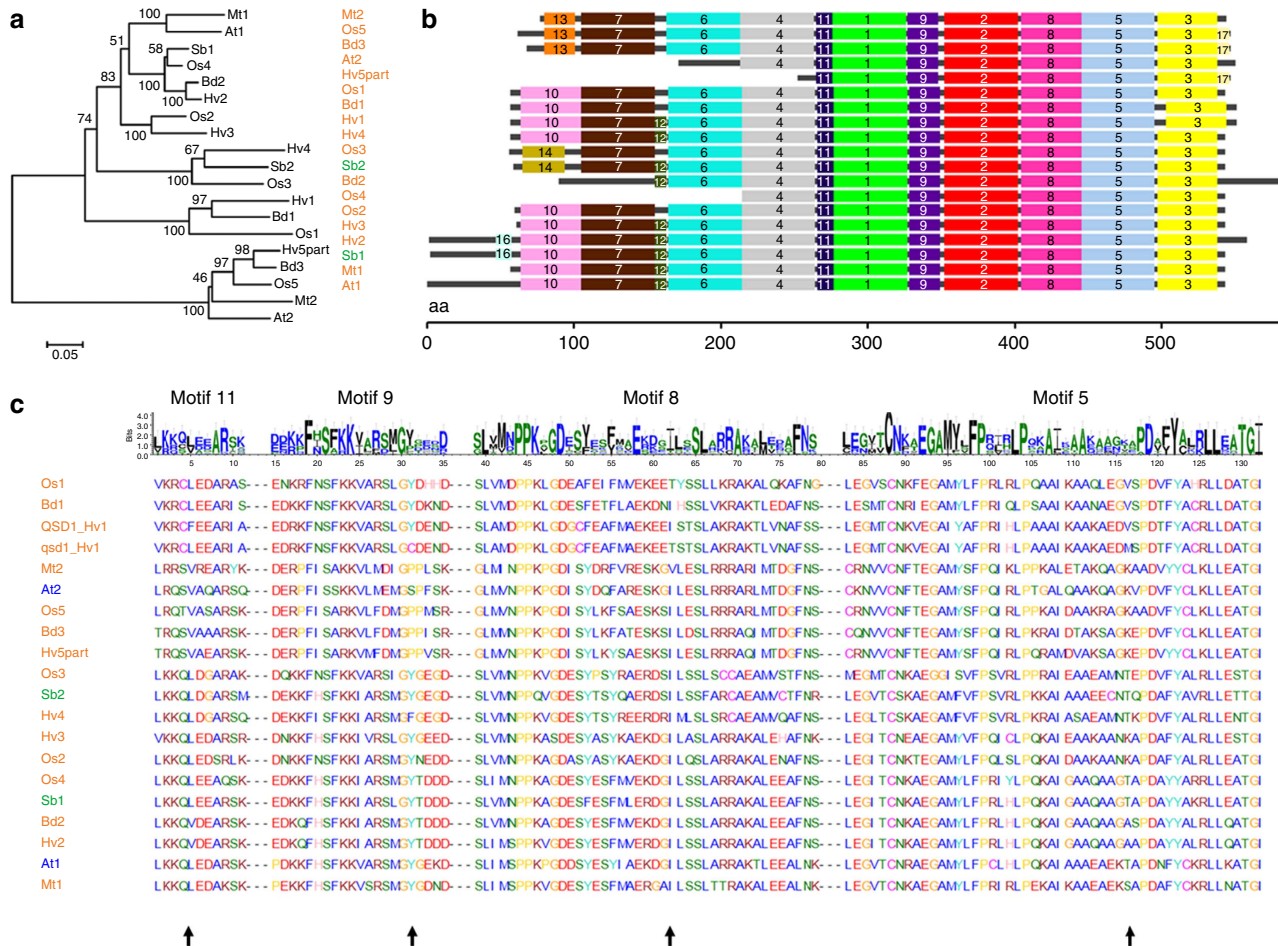

**Figure 4 | Alignment of QSD1 homologues in plant species. (a)** Multiple alignment of QSD1 amino acid sequences (Hv1: Haruna Nijo). Sequences are listed from accessions showing similarity to QSD1 ($E$-value $< E - 145$) by the blastp analysis in NCBI nr. Species and homologues with accession numbers are *Hordeum vulgare*: Hv1 (BAK04026.1), Hv2 (BAK07780.1), Hv3 (P52894.1), Hv4 (BAK05632.1), Hv5part (BAJ90574.1); *Oryza sativa*: Os1 (NP_001063248.1), Os2 (NP_001064504.1) Os3 (NP_001064505.2), Os4 (NP_001060284.1), Os5 (NP_001058716.1): *Sorghum bicolor*: Sb1 (XP_002463187.1), Sb2 (XP_002467302.1), *Brachypodium distachyon*: Bd1 (XP_003578159.1), Bd2 (XP_010235387.1), Bd3 (XP_003557680.1), *Arabidopsis thaliana*: At1 (NP_173173.3), At2 (AAK68842.1), *Medicago truncatula*: Mt1 (XP_003627448.1), Mt2 (XP_003613139.1). Annotation information is available in Supplementary Table 7. Scale bar indicates Poisson Correction distance. **(b)** Motif analysis of *Qsd1* homologues using SALAD database[22]. **(c)** Multiple alignment of motifs which show non-synonymous substitutions between QSD1 (Hv1: Haruna Nijo) and qsd1 (Hv1: H602). Arrows indicate positions of amino-acid substitutions located in motifs 11 (exon9), 9 (exon11), 8 (exon13) and 5 (exon14).

in the C1 clade, only haplotype C353 shares the SNP allele G at exon 9 with C_HN. The dominant *Qsd1* allele is carried by the cultivars with C_HN and C353 haplotypes, indicating that the dominant mutation in the *Qsd1* caused the reduced dormancy. The C_HN haplotype is distributed mainly in the Mediterranean and European areas, but is also found in elite barleys in Japan and Korea (Supplementary Data 1). The C353 haplotype is found only in European pedigrees of malting barleys. On the basis of the substitution at the E9 SNP, WC398 is a direct ancestral descendant of haplotype C_HN and a Palestinian wild barley; the latter could be the wild ancestral type of *Qsd1*. Another Palestinian wild haplotype, W0303, was also located in the C1 clade. Although it is not possible to exclude minor cultivars of clade C as candidate immediate ancestors, the dominant *Qsd1* allele appears to have originated from the *qsd1* allele of domesticated barley, rather than directly from wild barley. This notion is consistent with the conclusion that *Qsd1* was not responsible for barley domestication, but that its importance for the transition of barley from a primary food source to a useful raw material for malting and beer production was quickly recognized by farmers and early barley breeders, soon after the

domestication events. The large C1 clade involves wild barleys not only from Palestine (WC398), but also from Lebanon (W1585) and Jordan (W1215). The result indicates that the *Qsd1* allele is in the lineage from wild barleys of the southern Levant, where the first barley domestication also occurred[1,2].

It is likely that the activity of AlaAT in the embryo of grains will influence the length of dormancy at the biochemical level[25]. In plants, AlaAT is a pivotal enzyme in several key metabolic pathways that involve nitrogen assimilation, carbon metabolism and protein synthesis through its reversible transfer of an amino group from glutamate to pyruvate, to form oxaloacetate and alanine[26]. Certain isoenzymes can act as glutamate-glyoxylate amino transferases[27] and many of the enzymes can transfer amino groups from other amino acids, at lower rates. AlaAT has also been implicated in stress responses, in anaerobic glycolysis, in branched chain amino acid synthesis[28] and in nitrogen use efficiency[24,29]. In barley, amino acids are necessary for the synthesis of cellular proteins and secreted hydrolytic enzymes after the initiation of germination[30] and it is possible that AlaAT plays a necessary and central role in linking nitrogen and carbon metabolism with protein synthesis as sugars and amino acids are

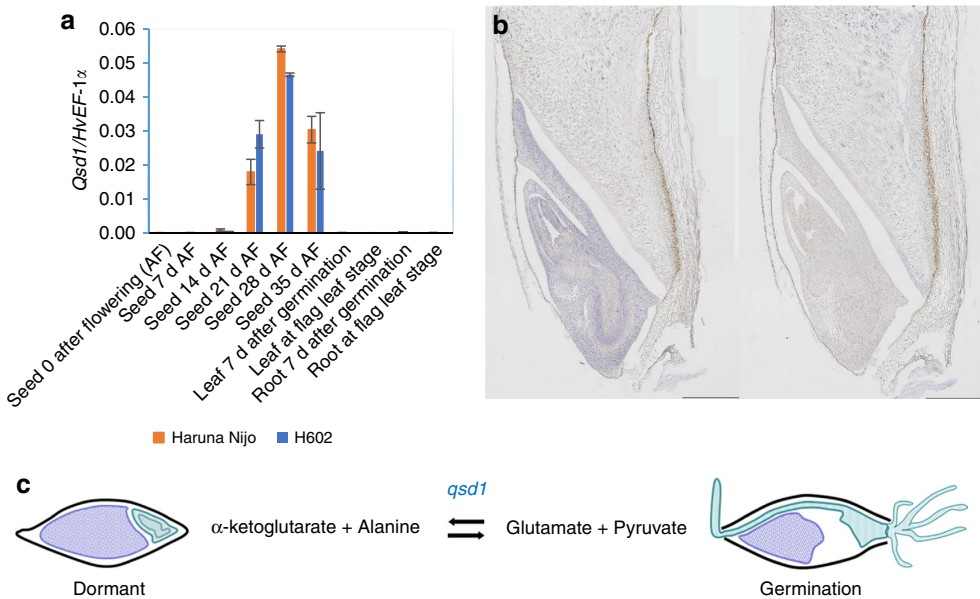

**Figure 5 | Expression of *Qsd1* and an estimated model of seed germination.** (**a**) Relative transcript levels of copy numbers on *Qsd1/EF-1α* (Supplementary Fig. 10) in different organs and growth stages in the less-dormant cultivar Haruna Nijo and dormant wild barley H602 (for primer sequences, see Supplementary Table 8). Difference of averages between Haruna Nijo and H602 is not significantly different (*t*-test, $P > 0.05$) in every pair of samples. Error bars represent s.e., $n = 3$. (**b**) *Qsd1* expression in an embryo of 19 days after flowering in H602 which is detected by *in situ* hybridization with an anti-sense (left) and sense (right) *Qsd1* 3′-UTR transcript. Scale bars represent 500 μm. (**c**) A suggested model of seed dormancy controlled by *qsd1* (derived from H602). Possible role of *qsd1* is in the building up of alanine in the dormant grain. Other constitutive alanine aminotransferases catalyse α-ketoglutarate + alanine ↔ glutamate + pyruvate pathway ubiquitously. Embryo is indicated in green.

released from storage proteins and starch during the germination process. The barley AlaAT encoded by the *qsd1* gene is located almost exclusively in the embryo (Fig. 5b), where one can envisage not only that the enzyme needs to be active in order for germination to proceed, but also that it could be a key target enzyme for the control of metabolic flux through the nitrogen and carbon metabolism pathways.

As shown in Fig. 5a, *Qsd1* and *qsd1* are expressed at similar levels but differ by a non-synonymous amino acid change on the external face of the protein that is linked with the alternative phenotypes. Knockdown of the *Qsd1* allele in cv. Golden Promise (Fig. 2c) causes a reduction in *Qsd1* expression and an increase in the dormancy level. We suggest that this is due to the lower abundance of *Qsd1* in these lines leading to lower AlaAT activity (Fig. 2a). However, complementation of *qsd1* by overexpression of *Qsd1*, led to a significant increase in germination (Fig. 2d). In these plants the total *Qsd1* expression level is increased (considering both alleles together; Fig 2f). However because the non-transgenic control and the transgenic negative plants still showed reasonable levels of *qsd1* expression, we believe that it is the specific expression of the *Qsd1* allele (that is, the difference between the proteins derived from the non-synonymous amino acid change) rather than the cumulative increase in expression that causes different levels of dormancy.

As mentioned above, dormancy and germination are complex processes and dormancy has been linked with redox signalling by reactive oxygen species[31], levels of the phytohormones GA, ethylene and ABA, and anoxic conditions[30]. Indeed, alanine accumulates during periods of hypoxia, as do some isoforms of the AlaAT[32,33], while some isoforms of AlaAT are involved in the conversion of alanine to pyruvate during recovery from hypoxia[20]. Thus, AlaAT isoenzymes might be important both in the accumulation of alanine in hypoxic conditions and in its removal during the recovery from hypoxia (Fig. 5c). Low oxygen potentials are observed in developing grains and seeds from

several species and could result in the accumulation of alanine[34]. It is known that the hard seed coat of legumes prevents germination by restricting the supply of oxygen in the embryo. The yellow seed coat locus ECY1 in *Brassica napus* also controls dormancy and has been implicated in alanine metabolism[35]. This is consistent with the use of hydrogen peroxide ($H_2O_2$) to break dormancy in barley seeds[36] but little is known about the oxygen status of mature barley grain and whether the release of low oxygen levels is an important early event in germination or whether the various isoforms of barley AlaAT play a direct role in breaking dormancy is not clear at this stage.

Despite these interpretative constraints, it is clear from the genetic data presented here that AlaAT is centrally important in the release of dormancy in barley grain. In practical terms, our data suggest that the *qsd1* gene for long dormancy appears to be a mutable gene that might be used for adaptation to different environments and this is consistent with the large variation in *qsd1* sequence that exists in nature. For example, the long dormancy *qsd1* gene could be used to control pre-harvest sprouting in higher rainfall areas to enhance global adaptation of barley. On the other hand, certain haplotypes which carry the *Qsd1* mutation are associated with barley lines that have been developed for industrial uses, such as in the malting and brewing industries. The *Qsd1* mutation that was discovered soon after barley domestication has proved to be essential for the transition of barley utilization from food to beverage in human diets and contributes further to the debate as to what extent the development of ancient agrarian societies was driven by the human appetite for flour and bread, or for beer and alcohol[37].

## Methods

**Scoring of seed dormancy.** Spike samples are harvested at physiological maturity (when the colour of first internode turns to yellow). Spikes are dried 48 h by the dehumidifier at the condition of 30 °C and 10% RH, and then stored at −20 °C until use. Fifty seeds of each sample were exposed to 25 °C for dormancy reduction

treatment and germinated for 4 days on moistened filter paper in Petri dishes at 25 °C. Seed non-dormancy was scored as the percentage of germinated seeds. Each germination test was replicated twice.

**Population development and genetic mapping.** *Qsd1* was initially mapped with the 93 doubled haploid phenotype data on a map derived from the cross between the cultivar Haruna Nijo and the wild barley line H602 consisting of 2,890 EST markers[13]. Of the four short dormancy QTLs, *Qsd1* mapped at the centromeric region of the long arm of chromosome 5H. The method of population development is described in Sato *et al.*[14]. In brief, $BC_3F_1$ recombinant chromosome substitution lines (RCSLs) derived from the cross between H602 (donor parent) and Haruna Nijo (recurrent parent) were genotyped and plants heterozygous for a chromosome segment with *Qsd1* were selected. A RSCL carrying the long dormancy allele *qsd1* and short dormancy alleles for other three QTLs was identified[13,14]. Using the initial back-crossed population of this line, a total of 910 $F_2$ plants and 4,792 $F_3$ plants derived from the heterozygous $F_2$ plants were produced to genetically map *Qsd1* (ref. 14). The segregation of the plants for seed germination (%) fitted a mono-factorial 1:2:1 Mendelian ratio (Supplementary Fig. 8). The direction of dominance was towards reduced dormancy (allele of Haruna Nijo) and the degree of dominance was 0.93, indicating that *Qsd1* shows nearly complete dominance[14]. *Qsd1/qsd1* genotype was determined during 35 days of 25 °C dormancy reduction treatment (Supplementary Fig. 9). Because the heterozygotes and non-dormant homozygotes were not clearly separated (Supplementary Fig. 8), the plants showing intermediate dormancy levels were further scored for segregation in $F_4$ to estimate the genotype of *Qsd1* in $F_3$. *EST1* and *EST12* markers (for primer sequences, see Supplementary Table 1), which flanked *Qsd1*, were used to select recombinants. The alignment of these two marker sequences with the rice genome revealed a segment carrying 26 rice genes (IRGSP/RAP build 5) that showed high sequence co-linearity with the other 10 barley genes (*EST2* to *EST11*). By scoring dormancy (Supplementary Figs 8 and 9), *Qsd1* was narrowed down to the region of three EST markers (*EST4F*, *EST4R* and *EST5*). A co-segregated marker with an estimated genotype of *Qsd1* was identified in two plants from the $F_3$ population. From each of these two plants, 18 $F_4$ plants were used to confirm the respective genotype of $F_3$ by segregation in the progeny.

**BAC sequencing and sequence annotation.** The primer set of co-segregated markers with *Qsd1* was used to select BAC clones of the less-dormant parent Haruna Nijo[15] and the dormant parent H602 (ref. 2). The BAC library of H602 was developed by custom order (Amplicon Express Co.). The inserts of H602 were cloned into the vector pCC1 (Epicenter vector in Invitrogen DH10B T1 resistant) in a *Hind*III cloning site. The average insert size was 180 kb and the total clone number was 172,800 (6 × ). From the BAC library of Haruna Nijo[14], a clone (GenBank ID: LC054174) was sequenced and a contig of 117,759 bp containing the marker sequence was obtained. Another BAC library of H602 was also screened with the *EST4R* marker and a contig sequence of 121,395 bp was obtained from another BAC clone (GenBank ID: LC054175). Each shotgun library from the selected clone was sequenced using a 3730xl sequencer (Applied Biosystems). Reads were assembled by phred/phrap software into contigs. The EST sequence was mapped on contig sequences to confirm the target sequence on the BAC clone. The Rice Genome Automated Annotation System (http://ricegaas.dna.affrc.go.jp/) was used for gene prediction on the contig sequence. Sequence polymorphisms between Haruna Nijo and H602 were detected by the alignment of contig sequences by CLUSTALX (http://www.clustal.org/).

**Development of knockdown transgenic plants.** A 509-bp cDNA region from AK372829 was amplified using the forward primer 5′-caccaaagctaggagatgg-3′ and reverse primer 5′-gaacacactgccccaaaagt-3′and cloned into the pENTR/D topo vector (Invitrogen). The construct was recombined (according to the manufacturer's instructions) into the binary vector pANDA modified to contain a GatewayTM cassette down-stream of the maize Ubi-1 promoter[18]. The construct was introduced into the *Agrobacterium* strain EHA101. Immature embryos of cv. Golden Promise were used for *Agrobacterium*-mediated transformation. Transformation was performed by the protocol of Hensel *et al.*[38]. In addition, pre-treatment with heat for immature embryos[39] and the dividing method for immature embryos before selection with hygromycin[40] were employed for more efficient transformation. The $T_0$ plants which had single copy inserts, as shown by Southern analysis, were examined further. We obtained several independent transgenic lines, and two (ids. #6 and #7) were used for further analysis. Genotypes of $T_1$ plants were determined by the inserts of segregating $T_2$ progeny and each $T_2$ plant from the $T_1$ plants were further genotyped by segregation in the $T_3$ progeny.

**Development of complementation transgenic plants.** The recombinant chromosome substitution line was used as a donor of dormant *qsd1/qsd1* allele in the genetic background of cv. Golden Promise. Twenty-three $B_1F_1$ plants were genotyped by the Golden Gate Assay (Illumina Co.), using selected 384 genome-wide SNP markers from barley OPA1 (ref. 41) to identify the substituted segment from H602 (Supplementary Fig. 3). Selected $B_1F_1$ plants with the substituted segment were self-pollinated to generate homozygous *qsd1/qsd1* substitution lines. *Agrobacterium*-mediated transformation of the substituted line (id. B1H602GP20-7),

using a vector construct carrying a cloned Haruna Nijo *Qsd1* genomic DNA (Supplementary Fig. 4), was conducted to complement the *qsd1* allele of the recipient line. The *Qsd1* open reading frame was amplified using the forward primer 5′-cccgggctcctctgcgccaccttagc-3′ and the reverse primer 5′-ggtacccagcgccac cagagttgaga-3′ and KOD -Plus- DNA polymerase from Haruna Nijo genomic DNA. This fragment was cloned into the pTA vector (TOYOBO) and sequenced. The open reading frame region was digested by *Sac*I and *Hind*III and ligated into binary vector pBUH3 (ref. 40, provided by Dr Y. Ito) and driven by the maize Ubi-1 promoter. We obtained several independent $T_0$ transgenic plants. Segregating $T_1$ plants (ids #1 and #4) were produced from single $T_0$ plants and further genotyped for the *Qsd1* transgene in the $T_2$ progeny.

**Phenotypic analysis of transgenic barley.** To measure dormancy levels, trans-genic barley lines were grown in a growth chamber with 12 h day length and at 15 °C. Expression levels were measured with the same method described below. Dormancy levels were measured at the time when the difference of long dormancy and short dormancy samples were maximized.

**AlaAT homology model.** The barley AlaAT amino acid sequences were aligned to the barley AlaAT crystal structure[24] and manually edited to ensure alignment of homologous catalytic motifs. Gap regions longer than 11 residues were modelled *de novo* using I-TASSER[42]. Modeller[43] was used to construct homology models of AlaAT using the barley AlaAT[24] and *de novo* predictions as templates. Modeller DOPE functions, ProSA[44] and Procheck[45] were used to assess the reliability of candidate models. High-scoring models were passed through a Modeller loop refinement script and final models chosen using Modeller DOPE, Modeller GA32, ProSA and Procheck assessments.

**Plant materials for allelic diversity analysis.** A total of 365 barley accessions preserved at Okayama University were evaluated for seed dormancy for two growing seasons. Nucleotide polymorphisms were detected by sequencing PCR amplicons from target regions. Primer sequences are shown in Supplementary Table 5.

**Phylogenic analysis of *Qsd1*.** The genomic regions of *Qsd1* in 27 wild and 210 cultivated barley accessions are sequenced. Non-redundant haplotypes were collected and aligned according to their similarities with the haplotypes of H602 (Supplementary Table 6). The similarities of 54 sequenced haplotypes were calculated by the neighbour joining method of MEGA6 (ref. 46).

**Expression analyses of target genes.** Transcript levels of Haruna Nijo and H602 were evaluated. Samples from seedling root, seedling leaf, adult root, adult leaf and maturing seeds (ovules) at 0, 1, 2, 3, 4 and 5 weeks after flowering were measured for *Qsd1/qsd1* transcripts and replicated three times. The quantitative PCR analysis was conducted using THUNDERBIRD SYBR qPCR Mix (TOYOBO Co.) on a Step One thermal cycler (Applied Biosystems Co.) with initial conditions of 95 °C, 60 s, 40 cycles of denaturing at 95 °C, 60 s and extension at 60 °C for 60 s. The *qsd1* gene segment comprising part of the 3′-UTR (300 bp) was amplified from cDNA isolated from immature barley spike using specific primers (Supplementary Table 8). The PCR product was cloned into the pBluescript IIKS ( + ) vector (Stratagene, La Jolla, CA, USA). Two clones with different insert orientations were linearized with *Not*I or *Eco*RI and were used as templates to generate antisense and sense probes using T3 or T7 RNA polymerase. *In situ* hybridization based on a digoxigenin-labelled RNA probe and subsequent immunological detection followed methods described elsewhere[47].

**Data availability.** BAC clone sequences described in this paper have been deposited in the DDBJ nucleotide database with accession codes LC054174: cv. Haruna Nijo BAC clone HNB550I12 and LC054175: wild barley H602 BAC clone HSP216G11. The data that support the findings of this study are available from the corresponding author upon request.

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

## Acknowledgements

We thank Ms N. Ooe and Y. Motoi for their excellent technical assistance, Drs S. Nakamura, M. Pourkheirandish, S. Toki for providing various analytical methods and Dr S. Ito for providing the pBUH3 vector for transformation. Barley seed samples and the Haruna Nijo BAC library were provided through the National Bioresource Project of Barley, MEXT of Japan. We acknowledge financial supports given by the Ministry of Agriculture, Forestry and Fisheries of Japan GD 3004, TRC1002, TRG1002 and TRS1001 to K.S., T.M. and T.K., and by the Japan Science and Technology Agency (CREST) to K.S. and K.T.

## Author contributions

K.S., T.K. and T.M. designed research; H.K., A.T., N.Y., M.Y. and K.T. performed research; K.S., T.K. and J.G.S. analysed data; K.S., G.B.F. and T.K. wrote the paper.

## Additional information

**Competing financial interests:** The authors declare no competing financial interests.

