## [Peer Review File · Nature Communications]

Reviewer #1 (Remarks to the Author):

As far as I am concerned the authors have addressed the majority of the points I raised. A couple of small items still need to be resolved - but mainly for clarity for the reader. My thoughts are given below:

Lines 13-14. There is a break in logic here that needs to be addressed. First sentence in the abstract stresses the long dormant allele being important for surviving the long dry summers in the fertile crescent and short dormant alleles were selected in agriculture. The last sentence stresses use of the dormant allele to control pre-harvest sprouting in high rainfall areas. It appears from their data that a range of alleles with less severe phenotypes exist and it may be these that are now being selected to protect from pre-harvest sprouting. Is this correct? I don't agree that this an adaptive response to global climate change - its happened over a much longer period that we have been aware of dramatic climate change. Put simply - I think a little work is still required on the text in the abstract to be crystal clear what is meant.

Line 37 delete: Another Major QTL

Line 218-220 'However, it is clear that the AlaAT expression is high in Qsd1 but lower in knock down plants in Fig. 2. This suggests that the lower activity of AlaAT may be responsible for the lower seed germination.'

This is right - but the way it is written is intuitively wrong and I think the text needs a little work, for the following reasons (as I think through this). First If as stated you knock down expression and observe the phenotype then the obvious conclusion is that its expression level - not activity - that is causal. However, earlier experiments showed that Qsd1 and qsd1 are expressed at the same level but differ by a non-synonymous amino acid change on the external face of the protein that genetically is linked with the alternative phenotypes - non-dormant and dormant respectively. Knocking down the Qsd1 allele in GP (Qsd1) causes a reduction in Qsd1 expression and parallel increase dormancy. Again this tells me that its Qsd1 expression that is controlling dormancy. However, complementing qsd1 with Qsd1 overexpression increases (~doubles) the recorded expression (of both alleles assuming both mRNAs are assayed) and leads to a significant increase in germination. The non transgenic control still has reasonable levels (50%) of qsd1 expression as do the null segregants. This now tells me it's specifically expression of the Qsd1 allele that is having the effect - i.e. this time it's the difference between the proteins (not expression per se) that is causal. So I agree with the conclusion - but the way it is said needs to be redrafted.

To tie this expression vs activity vs complex formation up completely I would still have liked to see the activities of the two alternative forms of the enzyme or some proof that the alternative alleles alter dimerization or regulatory protein binding. I am however sympathetic to the authors reasoning that this may be difficult, take some time and be inconclusive - and I am therefore content that they have at least explored this a little more in the text. Perhaps these experiments should be attempted for a follow-up?.

Reviewer #2 (Remarks to the Author):

The authors have satisfactorily addressed the majority of the reviewers comments. I feel the paper has now a better shape and has improved its rigour and accuracy. It will be a great contribution to the seed dormancy field.

Reply to the Reviewers' comments:

Reviewer #1 (Remarks to the Author):

As far as I am concerned the authors have addressed the majority of the points I raised. A couple of small items still need to be resolved - but mainly for clarity for the reader. My thoughts are given below:

Lines 13-14. There is a break in logic here that needs to be addressed. First sentence in the abstract stresses the long dormant allele being important for surviving the long dry summers in the Fertile Crescent and short dormant alleles were selected in agriculture. The last sentence stresses use of the dormant allele to control pre-harvest sprouting in high rainfall areas. It appears from their data that a range of alleles with less severe phenotypes exist and it may be these that are now being selected to protect from pre-harvest sprouting. Is this correct? I don't agree that this an adaptive response to global climate change - its happened over a much longer period that we have been aware of dramatic climate change. Put simply - I think a little work is still required on the text in the abstract to be crystal clear what is meant.

The Abstract is revised as follows:

“Dormancy allows wild barley grains to survive dry summers in the Near East. After domestication, barley was selected for shorter dormancy periods. Here we isolate the major seed dormancy gene *qsd1* from wild barley, which encodes an alanine aminotransferase (AlaAT). The seed dormancy gene is expressed specifically in the embryo. The AlaAT isoenzymes encoded by the long and short dormancy alleles differ in a single amino acid residue. The reduced dormancy allele *Qsd1* evolved from barleys that were first domesticated in the southern Levant and had the long dormancy *qsd1* allele that can be traced back to wild barleys. The reduced dormancy mutation likely contributed to the enhanced performance of barley in industrial applications such as beer and whisky production, which involve controlled germination. In contrast, the long dormancy allele might be used to control pre-harvest sprouting in higher rainfall areas to enhance global adaptation of barley.”

Line 37 delete: Another Major QTL

Deleted.

Line 218-220 'However, it is clear that the AlaAT expression is high in Qsd1 but lower in knock down plants in Fig. 2. This suggests that the lower activity of AlaAT may be responsible for the lower seed germination.'

This is right - but the way it is written is intuitively wrong and I think the text needs a little work, for the following reasons (as I think through this). First if as stated you knock down expression and observe the phenotype then the obvious conclusion is that its expression level - not activity - that is causal. However, earlier experiments showed that Qsd1 and qsd1 are expressed at the same level but differ by a non-synonymous amino acid change on the external face of the protein that genetically is linked with the alternative phenotypes - non-dormant and dormant respectively. Knocking down the Qsd1 allele in GP (Qsd1) causes a reduction in Qsd1 expression and parallel increase dormancy. Again this tells me that its Qsd1 expression that is controlling dormancy. However, complementing qsd1 with Qsd1 overexpression increases (~doubles) the recorded expression (of both alleles assuming both mRNAs are assayed) and leads to a significant increase in germination. The non-transgenic control still has reasonable levels (50%) of qsd1 expression as do the null segregants. This now tells me it's specifically expression of the Qsd1 allele that is having the effect - i.e. this time it's the difference between the proteins (not expression per se) that is causal. So I agree with the conclusion - but the way it is said needs to be redrafted.

To tie this expression vs activity vs complex formation up completely I would still have liked to see the activities of the two alternative forms of the enzyme or some proof that the alternative alleles alter dimerization or regulatory protein binding. I am however sympathetic to the authors reasoning that this may be difficult, take some time and be inconclusive - and I am therefore content that they have at least explored this a little more in the text. Perhaps these experiments should be attempted for a follow-up?

We really appreciate the comment and encouragement by the reviewer #1. We are planning to try the activity assays of the two alternative forms of the AlaAT by inviting specialists with this technique to collaborate in a follow-up project.

We have revised the current manuscript as follows (page 10, lines 237-247) according to the reviewer's suggestions:

“As shown in Fig. 5a, *Qsd1* and *qsd1* are expressed at similar levels but differ by a non-synonymous amino acid change on the external face of the protein that is linked with the alternative phenotypes. The knockdown experiment of *Qsd1* allele in cv. Golden Promise

(Fig. 2c) causes a reduction in *Qsd1* expression and an increase in the dormancy level, which suggests that the lower abundance of *Qsd1* may be responsible for the lower seed germination (Fig. 2a). However, complementation of *qsd1* with *Qsd1* overexpression, through which both mRNAs were measured, increased the expression of both alleles (Fig. 2f) and led to a significant increase in germination (Fig. 2d). The non-transgenic control and the transgenic negative plants still showed reasonable levels of *qsd1* expression, suggesting that the specific expression of *Qsd1* allele, i.e., the difference between the proteins derived from the non-synonymous amino acid change, causes different levels of dormancy.”

Reviewer #1 (Remarks to the Author)

This is a very nice piece of work that identifies a dominant mutation in an ancestral AlaAT enzyme as a major determinant of the switch between the observed long dormancy in the WT to low dormancy on derived domesticated and cultivated lines. I have no issues over the cloning and characterisation of the genic mutations other than minor ones (see below). In my opinion the work appears to have been impeccably conducted and leaves me without doubt that the authors have identified the causal gene and in most likelihood the causal mutation responsible for the switch in phenotype (with some inherent nuances). This work alone is probably worthy of publication in Nature comms and in itself is a highly significant finding. The less convincing section for me is the discussion around why this mutation has the observed effect and I am not entirely convinced that the authors have done all that is necessary to extend their argument towards how the mutation may exert its effect. For example firstly - they analysed expression of AlaAT over grain developmental time only in the ancestral alleles and not in the derived domesticated form. It could be that it is temporal expression differences that are responsible for the phenotype and that the possibly causal mutation is in high LD with a cis-effect on gene expression. Comparative data from both alleles could have solved this issue. Secondly rather than predict the effect of the L214F mutation on activity by looking at its position on the protein tertiary structure model (which is of course interesting), it may have been better to test the activity of the alternative alleles using activity assays. I believe colorimetric assays are available for AlaAT - Expressing both alleles in E-coli or an alternative expression system and assaying comparative activity may have shed more light on the question of 'how'. I would therefore suggest that the authors consider doing such experiments to really join the dots or tone down the whole discussion about how AlaAT could affect dormancy - its still relevant - but toned down. I would certainly recommend that they do the first of the experiments suggested above - ie measure comparative expression in WT and derived forms - not just WT - unless they have very strong arguments about why this shouldn't be done or is irrelevant. The bottom line is they have identified a gene and likely a mutation that confers a very important phenotype. This alone deserves to be published.

To address the specific NG questions: The work is original and novel, the data and methodology are sound (except perhaps for the two points raised above that could be added), Stats are OK, conclusions are solid, discussion around the conclusions more speculative (as expected), suggested improvements are mentioned above - expression analysis of the dominant AlaAT allele AND activity assays of both forms, References are fine (see comment about ref 5) and the paper is well written and generally clear.

Less important points:

line 28: Add a sentence to say that all EST markers were then assayed in the recombinants

Line 29: should mention that it was BAC clones that were identified from each library - and how many comprised a contig - 1? 2? more?

Line 44: add the word 'dominant' in front of '...short dormancy'...

line 47-49 remove 'that could be.....on seed germination'. Not required and not representative of what was done.

Line 56: Change to 'We also conducted complementation tests using.....by transforming with...'

Line 71: is ref 5 correct in this context? also please clarify is it not 'the parents of other mapping populations that had previously been used for Qsd1 mapping'

Line 84: should read 'Variation at the other three SNP alleles.....'

Line 96-97: How can you say that C353 originated directly from HN and not a common ancestor in the C1 clade?

Lines 101-103: Based on the information in Figure 4 I cannot see how WC398 is a direct ancestral descendant (progenitor??) of HN from the data presented in the cladogram. It looks like there are several more closely related individuals.

Lines 104-107: I'm not convinced that the data presented allows the authors to claim that the Ssd1 allele originated in domesticated barleys.

Lines 89-113 - the whole paragraph - including the previous three comments may just need tidied up a bit and made clear (in reference to their data)

Lines 119-120: change 'shows high p-values' for - are highly similar ($p > 0.XXXX$)

Line 131: the expression analysis refers to timing - not location as stated

From L130 my comments are covered in the general comments above.

Reviewer #2 (Remarks to the Author)

Summary of the results:

In this paper the authors investigate the major barley dormancy QTL Qsd1, which is located on Chr5H. The authors fine-mapped that QTL by using a population derived from a cross with a substitution line identified in a previous work, and found within the candidate QTL interval one single gene with non-synonymous SNPs. The candidate gene encoded an Alanine aminotransferase, which by the use of transgenic plants is shown to control grain dormancy. Golden Promise barley plants having a dominant low-dormancy allele, showed enhanced dormancy when transformed with an RNAi construct silencing the candidate gene. On the contrary, a substitution line carrying the recessive high-dormancy allele (from wild barley) showed decreased dormancy when transformed with the dominant low-dormancy allele. The finding of this gene is very exciting as is a new gene related with dormancy. After analysing gene haplotypes in a diversity barley panel the authors concluded that the low dormancy allele was acquired post-domestication, in relation with brewing activities.

Originality and interest:

The paper describes original work that is of great interest for seed scientist and that can have applications for modifying grain dormancy.

Data and methodology:

In my opinion the approach used is convincing and the data supports the conclusions. My only concern is that transgenic proof-of-function is only based in one single transgenic event for each experiment. Only the progeny of one transgenic plant is used. In the literature is commonly seen more than one transgenic plant analysed.

Conclusions:

The claim that the low-dormancy allele was selected for malting and brewing applications needs more explanation and is not clear to me that the reference number 5 gives any support about which pedigrees carry the low dormancy allele.

Suggested improvements:

-The paper needs detailed review for improving clarity and grammar accuracy. For example on line 22 the sentence "Furthermore, seed dormancy in wild barley is much longer than is these other species" does not make sense.

-I would recommended replacing the terms "long dormancy" and "short dormancy", by the more commonly used "strong dormancy" and "Weak dormancy", or "high dormancy" or "low dormancy". When the authors used long and short terms I got confused with the actual physical length of the alleles.

-No function for the identified gene is provided, although the authors speculate about putative

biochemical reactions. I just found a new report from 2016 about a link between Alanine and dormancy in Brassica that might deserve a comment (Embryonal Control of Yellow Seed Coat Locus ECY1 Is Related to Alanine and Phenylalanine Metabolism in the Seed Embryo of Brassica napus. Wang et al. doi: 10.1534/g3.116.027110).

References and previous work:

I think in the introduction the previous QTL work done in barley should be better explained. On Chr5H there are 2 major QTL for dormancy: SD1 and SD2. This should be introduced and then the authors can state that this paper is about SD1.

Reviewer #3 (Remarks to the Author)

This manuscript describes the genetic analysis of the segregation of a locus for seedling dormancy, which turns out to be an alanine aminotransferase. The MS is an extensive piece of work and uses a genetic approach, without having an preconceived notion of the particular gene. Therefore, the authors have used the most logical and stringent approach to determine the role of specific genes in seed dormancy in barley.

This MS also contains a significant amount of work and the authors have laid out the key issues that needed to be addressed. Ln 44 -46. For example, the authors point out that Golden Promise has the wrong dormancy allele, so that they had to make an RNA knock-down line, to address the specific question they raised. The experiments described are extensive, and address all of the issues that I would have raised. The one area where I would suggest receive a bit of editing is in the model describing how AlaAT may play a role in germination. It would be of interest for the authors to suggest why this AlaAT is so important, is it its expression pattern.

In summary, I believe this MS represent a novel and extensive piece of research and is worthy of publication in Nature Communications.

Reply to the Reviewers' comments:

Reviewer #1 (Remarks to the Author):

This is a very nice piece of work that identifies a dominant mutation in an ancestral AlaAT enzyme as a major determinant of the switch between the observed long dormancy in the WT to low dormancy on derived domesticated and cultivated lines. I have no issues over the cloning and characterisation of the genic mutations other than minor ones (see below). In my opinion the work appears to have been impeccably conducted and leaves me without doubt that the authors have identified the causal gene and in most likelihood the causal mutation responsible for the switch in phenotype (with some inherent nuances). This work alone is probably worthy of publication in Nature comms and in itself is a highly significant finding. The less convincing section for me is the discussion around why this mutation has the observed effect and I am not entirely convinced that the authors have done all that is necessary to extend their argument towards how the mutation may exert its effect. For example

Major comment 1: firstly - they analysed expression of AlaAT over grain developmental time only in the ancestral alleles and not in the derived domesticated form. It could be that it is temporal expression differences that are responsible for the phenotype and that the possibly causal mutation is in high LD with a cis-effect on gene expression. Comparative data from both alleles could have solved this issue.

The expression data for the less dormant haplotype (cv. Haruna Nijo) have been added to Fig. 5a, but no major temporal differences in transcript abundance can be seen. A sentence explaining this point has been added near line 154 on page 6

Major comment 2: Secondly rather than predict the effect of the L214F mutation on activity by looking at its position on the protein tertiary structure model (which is of course interesting), it may have been better to test the activity of the alternative alleles using activity assays. I believe colorimetric assays are available for AlaAT - Expressing both alleles in E-coli or an alternative expression system and assaying comparative activity may have shed more light on the question of 'how'. I would therefore suggest that the authors consider doing such experiments to really join the dots or tone down the whole discussion about how AlaAT could affect dormancy - its still relevant - but toned down. I would certainly recommend that they do the first of the experiments suggested above - ie measure comparative expression in WT and derived forms - not just WT - unless they have very strong arguments about why this shouldn't be done or is irrelevant. The bottom line is they have identified a gene and likely a mutation that confers a very important phenotype. This alone deserves to be published.

The co-authors have discussed this suggestion of comparing the activities of the two isoforms of AlaAT in in vitro systems at great length. We feel that the L214F substitution is on the surface of the protein and some distance from the active site. One wonders therefore if differences in activity will be detectable. We think it is more likely that this residue is involved in the formation of enzyme homo- or hetero-dimers or -trimers, because it is on the surface of the enzyme. Both amino acids (L and F) are hydrophobic. We should remember too that all the commonly used heterologous systems, such as *E. coli*, yeast and *Nicotiana benthamiana*, are likely to have high background levels of AlaAT, which is a key enzyme in central metabolic pathways. Considering all these limitations, we feel that the expression work would be very time-consuming and in all likelihood would not generate interpretable results. Instead, we have added data from another RNAi event in Fig. 2, where it is clear that the AlaAT expression is high in *Qsd1* but much lower in the knock down plants. While we acknowledge that transcript levels are not necessarily translated into lower levels of the enzyme or its activity, we believe it is likely that it is the lower activity of AlaAT that may cause lower seed germination, but the mechanism of this clearly complex control system remain speculative at the moment. We have strengthened these points in the text (lines 169-171 on page 7) and in the legend to Fig. 2. We hope these explanations are acceptable.

Less important points:

Line 28: Add a sentence to say that all EST markers were then assayed in the recombinants

We have added the sentence "All 13 EST markers were assayed in the recombinants." on page 2, lines 56-57.

Line 29: should mention that it was BAC clones that were identified from each library - and how many comprised a contig - 1? 2? more?

Added text on page 2, lines 59-60 "One and three clones were selected for Haruna Nijo and H602 libraries, respectively. Of these, one clone from each library comprised a contig."

Line 44: add the word 'dominant' in front of '...short dormancy'...

Added on page 2, line 75

line 47-49 remove 'that could be.....on seed germination'. Not required and not representative of what was done.

Removed.

Line 56: Change to 'We also conducted complementation tests using.....by transforming with...'

Revised the sentence on page 3 from line 84 as follows: “We also conducted complementation tests using genetic substitution lines that contained the recessive long dormancy qsd1 allele of wild barley H602 (Extended Data Fig. 3) in a background of Golden Promise (id. B1H602GP20-7: qsd1/qsd1) by transforming with a construct carrying the cloned Haruna Nijo short dormancy Qsd1 genomic DNA (Extended Data Fig. 4).”

Line 71: is ref 5 correct in this context? also please clarify is it not 'the parents of other mapping populations that had previously been used for Qsd1 mapping'

Corrected as ref 9 on page 4, line 100.

Line 84: should read 'Variation at the other three SNP alleles.....'

Revised as “Variation at the other three SNP alleles showed relatively smaller effects on germination.” on page 5, line 113.

Line 96-97: How can you say that C353 originated directly from HN and not a common ancestor in the C1 clade?

Revised as “Of the haplotypes observed in the C1 clade, only haplotype C353 shares the SNP allele G at exon 9 with C_HN.” on page 8, lines 185-186.

Lines 101-103: Based on the information in Figure 4, I cannot see how WC398 is a direct ancestral descendant (progenitor??) of HN from the data presented in the cladogram. It looks like there are several more closely related individuals.

Capital “W” represents the wild ancestral form of capital “C” cultivated barley. Since WC398 contain 1 wild and 54 cultivated barleys and shares the same haplotype, we assume that a wild barley in the haplotype WC398 could be the direct ancestral wild haplotype in C1 clade. This nomenclature is now clarified on page 5, lines 121-122.

Lines 104-107: I'm not convinced that the data presented allows the authors to claim that the Qsd1 allele originated in domesticated barleys.

We find the SNP allele G at exon 9 in wild barleys, but no wild barleys examined in this study had this allele. At least this G allele has been historically inherited in malting barley cultivars e.g. Haruna Nijo, Morex, Golden Promise, Harubin 2-row in Supplementary Table 2. More information for SNP haplotypes are added in Supplementary Table 7.

Lines 89-113 - the whole paragraph - including the previous three comments may just need tidied up a bit and made clear (in reference to their data)

This paragraph is improved with the definitions of W, C and WC on page 5, lines 121-122.

Lines 119-120: change 'shows high p-values' for - are highly similar ($p > 0.XXXX$)

Revised as “are highly similar (E-value < E^{-145})” on page 6, line 130.

Line 131: the expression analysis refers to timing - not location as stated.

Revised as “transcriptional profiling was used to first define the **timing of the enzyme in the grain.” on page 6, line 142.**

From L130 my comments are covered in the general comments above.

See comments above..

Reviewer #2 (Remarks to the Author):

Data and methodology:

In my opinion the approach used is convincing and the data supports the conclusions. My only concern is that transgenic proof-of-function is only based in one single transgenic event for each experiment. Only the progeny of one transgenic plant is used. In the literature is commonly seen more than one transgenic plant analysed.

We have added data for another event for the transgenic experiments and a description in the legend of Fig. 2.

Conclusions:

The claim that the low-dormancy allele was selected for malting and brewing applications needs more explanation and is not clear to me that the reference number 5 gives any support about which pedigrees carry the low dormancy allele.

Our mistake, we have changed the reference number from 5 to 6. Supplementary Table 7 has been added to provide detailed accession information.

Suggested improvements:

-The paper needs detailed review for improving clarity and grammar accuracy. For example on line 22 the sentence "Furthermore, seed dormancy in wild barley is much longer than in these other species" does not make sense.

We have revised this sentence as follows: "Furthermore, seed dormancy in wild barley is much longer than seed dormancy observed in these other cereal species." on page 2 lines 41-42.

-I would recommended replacing the terms "long dormancy" and "short dormancy", by the more commonly used "strong dormancy" and "weak dormancy", or "high dormancy" or "low dormancy". When the authors used long and short terms I got confused with the actual physical length of the alleles.

We have somewhat of a dilemma here. While reviewer 2 would prefer 'strong' and 'weak', the other reviewers do not comment on our nomenclature. In choosing 'short' and 'long' we took account of the fact that the differences in dormancy were measured in terms of time, and time is generally expressed as short and long, rather than high and low, or strong and weak. We would prefer to leave the terms as they are and hope that this is acceptable.

-No function for the identified gene is provided, although the authors speculate about putative biochemical reactions. I just found a new report from 2016 about a link between Alanine and dormancy

in Brassica that might deserve a comment (Embryonal Control of Yellow Seed Coat Locus ECY1 Is Related to Alanine and Phenylalanine Metabolism in the Seed Embryo of Brassica napus. Wang et al. doi: 10.1534/g3.116.027110).

Wang et al. (2016) is now added to the reference list and the text has been modified on page 9, lines 230-231 to include this interesting finding.

References and previous work:

I think in the introduction the previous QTL work done in barley should be better explained. On Chr5H there are 2 major QTL for dormancy: SD1 and SD2. This should be introduced and then the authors can state that this paper is about SD1.

Han et al. (1996) and Nakamura et al. (2016) have been added to the reference list. The text is modified on page 2, lines 34-37.

Reviewer #3 (Remarks to the Author):

Comment: The one area where I would suggest receive a bit of editing is in the model describing how AlaAT may play a role in germination. It would be of interest for the authors to suggest why this AlaAT is so important, is it its expression pattern.

We have discussed the potential role of AlaAT in the paper, but were mindful that the mechanism for its action in the determination of dormancy has not been defined and that we could only speculate on the actual mechanism. We could expand this part of the discussion, but feel that the present level of speculation is probably about right.